# Stagnant forearc mantle wedge inferred from mapping of shear-wave anisotropy using S-net seafloor seismometers

Naoki Uchida [1]✉, Junichi Nakajima [2], Kelin Wang [3], Ryota Takagi [1], Keisuke Yoshida[1], Takashi Nakayama[1], Ryota Hino[1], Tomomi Okada[1] & Youichi Asano[4]

Shear-wave anisotropy in Earth's mantle helps constrain the lattice-preferred orientation of anisotropic minerals due to viscous flow. Previous studies at the Japan Trench subduction zone using land-based seismic networks identified strong anisotropy in the mantle wedge, reflecting viscous flow induced by the subducting slab. Here we map anisotropy in the previously uninvestigated offshore region by analyzing shear waves from interplate earthquakes that are recorded by a new seafloor network (the S-net). The newly detected anisotropy is not in the mantle wedge but only in the overlying crust (~0.1 s time delay and trench-parallel fast direction). The distinct lack of anisotropy indicates that the forearc mantle wedge offshore is decoupled from the slab and does not participate in the viscous flow, in sharp contrast with the rest of the mantle wedge. A stagnant forearc mantle wedge provides a stable and cold tectonic environment that is important for the petrological evolution and earthquake processes of subduction zones.

[1] Graduate School of Science, Tohoku University, 6-6, Aramaki-aza-aoba, Aoba-ku, Sendai 980-8578, Japan. [2] Department of Earth and Planetary Sciences, School of Science, Tokyo Institute of Technology, 2-12-1, Ookayama, Meguro-ku, Tokyo 152-8551, Japan. [3] Pacific Geoscience Centre, Geological Survey of Canada, 9860 West Saanich Road, Sidney, BC V8L 4B2, Canada. [4] National Research Institute for Earth Science and Disaster Resilience, 3-1 Tennodai, Tsukuba, Ibaraki 305-0006, Japan. ✉email: naoki.uchida.b6@tohoku.ac.jp

S hear-wave splitting is a phenomenon in which a shear wave traveling in a medium with anisotropic elastic moduli splits into two polarized components, with one traveling faster than the other. Measurements of the polarization of the faster shear-wave (the fast direction) and the arrival-time difference between the fast and slow components (the delay time) help to delineate anisotropy along the ray path[1–3]. In understanding subduction zone dynamics, it is important to know the pattern of solid flow within the mantle wedge between the upper plate and subducting slab, and seismic anisotropy is an excellent indicator of the flow pattern. At present, the anisotropy of the offshore part of the forearc mantle wedge is essentially unknown because of the lack of seafloor seismic observations.

The present knowledge of forearc shear-wave anisotropy is based on limited observations from the land area[3–8]. In a number of subduction zones, such as New Zealand[4], Cascadia[5,6], Sumatra[7] and central South America[8], these limited observations suggest that the fast direction in onshore forearc from local earthquakes is predominantly trench-parallel. The fast directions from teleseismic events are also trench-parallel in some subduction zones[9]. From these results, it is almost impossible to infer the anisotropy state of the forearc mantle wedge with confidence. Many studies assume the source of the observed anisotropy to be outside of the mantle wedge, either beneath the subducting slab associated with trench-parallel mantle flow[10], within the slab associated with structural fabrics acquired before subduction[11] or upon subduction due to plate bending[9], or within the overlying crust associated with stress-controlled preferred orientation of microcracks or geological fabrics[6,12,13]. Those that consider the anisotropy to be within the forearc mantle wedge often associate it with assumed abundance of B-type olivine minerals that are aligned by the mantle wedge corner flow[14]. The remarkably poor knowledge of mantle wedge anisotropy is to a large part due to limitations in the spatial coverage of the observations and the vertical resolution of the anisotropy analysis.

In the northeastern (NE) Japan subduction zone, trench-normal and trench-parallel fast directions have been inferred for the back-arc and forearc areas, respectively, from waveforms of local earthquakes within the subducting Pacific slab[15–18] (Fig. 1). In the forearc, however, the observations are mostly limited to the narrow land area stretching only ~50 km from the volcanic front to the coast (Fig. 1a and c). It is unknown whether the results represent the entire forearc which is mostly offshore. It is also unresolved whether the source of the trench-parallel fast direction is uniform in depth beneath the forearc.

Recently, the Seafloor Observation Network for Earthquakes and Tsunamis along the Japan Trench (S-net) was established off NE Japan by the National Research Institute for Earth Science and Disaster Resilience (NIED)[19]. The deployment of the cable system began in 2013 and was completed in 2017, and the data were made publicly available from October 2018 onward[20–22]. The new system covers a subsea area of about 300 × 1000 km with 150 ocean bottom seismometers (OBSs) connected by a 5,800 km long fiberoptic cable (Fig. 1a). Seismic records from these instruments not only help understand seismicity and megathrust slip in the shallow subduction zone[23–25], but also expand the study of mantle wedge shear-wave anisotropy into a vast virgin territory. In the following, we will document the densest systematic mapping of the shear-wave anisotropy of an offshore forearc to date, and we will demonstrate that the results indicate a lack of anisotropy in the offshore forearc mantle wedge and, in conjunction with previous findings based on onshore networks, provide a clear and simple picture of mantle wedge dynamics. Given the consistency of our results with geodynamic models of subduction zones, the knowledge learned in this study is expected to be globally applicable.

## Results

**Selection of data for inferring mantle wedge dynamics.** We use waveform data recorded by the S-net OBSs from August 2016 to April 2019 (Fig. 1a, c). Because of the mantle wedge focus of this work, we ensure that the wave paths sample only the forearc rocks above the subducting slab, so that the resolved anisotropy is not within or beneath the slab. Therefore, we do not use teleseismic waveforms, and we use only local earthquakes along the subduction interface and, where available, in the upper plate including the cold nose of the forearc mantle wedge.

The S-net seafloor seismometers are not yet used for routine hypocenter location, and therefore the depth determination of many offshore events has large uncertainties because of the small event depths compared to station separation. For offshore earthquakes, source depths from the Full Range Seismograph Network of Japan (F-net) catalog[26] which is based on waveform modeling are of better quality, and the focal mechanism information in the catalog is useful for selecting interplate events (see Supplementary Fig. 1 for the consistency of F-net depth and S-P time from S-net). We have selected 287 interplate earthquakes with Mw ≥ 3.5 based on their focal mechanisms while taking into account their depth information in the catalog of the F-net[26] (see Methods). In addition, we also use the 321 small repeating earthquakes (M ≥ 2.5) that have been identified to be located along the creeping parts of the subduction interface[27] (Fig. 1a), which greatly increases the number of available interplate earthquakes. Our selection of the repeating earthquakes is based on waveform similarity at land stations and represents an update of the catalog of Uchida and Matsuzawa [2013][28] (see Methods). The distribution of the interplate events including the repeaters shows a distinct gap in the rupture area of the 2011 Tohoku-oki earthquake (Mw 9.0)[29] (near-trench area of 37°–39° N), because few interplate events have occurred here since this great earthquake (Fig. 1a). We also use 108 shallow earthquakes (Mw ≥ 3.5) in the overlying plate which are selected on the basis of their depths (shallower than subduction interface or 35 km, whichever is shallower) and focal mechanisms according to the F-net catalog (Fig. 1a, see Methods). The different ray paths of the two types of earthquakes (interplate vs. upper-plate) are useful for determining whether the observed anisotropy is within the mantle wedge (Fig. 1c).

We use three-component 100-Hz-sampled waveforms obtained from the S-net OBSs with a natural frequency of 15 Hz. We rotate waveforms to the geographic directions (up, east, and north) based on sensor orientations[24] (see Methods). We use practically the same procedure employed by Nakajima et al. [2006][17], so that our estimated splitting parameters are fully comparable with those previously derived for the land area. We employ a 2–8 Hz band-pass filter for the horizontal components and visually identify and window the S phases for 1–1.5 cycles of the waveform oscillations. We use a cross-correlation method[30] to estimate the fast direction and delay time (Fig. 1b). To avoid contamination by surface conversions, we only use earthquakes within a 45° shear-wave window extending downward from each seismic station[31]. Owing to the upward concave bending of ray paths caused by the extremely low wave speed near the sea bottom in our study area, we are able to use this shear-wave window that is larger than the commonly used 35°. See Methods for details.

**Trench-parallel fast directions in the forearc crust.** As a result, we obtained 1400 and 264 fast directions from the shear waves of the interplate and upper-plate earthquakes, respectively, together with their delay times. For each station, we averaged the splitting parameters of the same type of earthquakes (Fig. 2b). The number

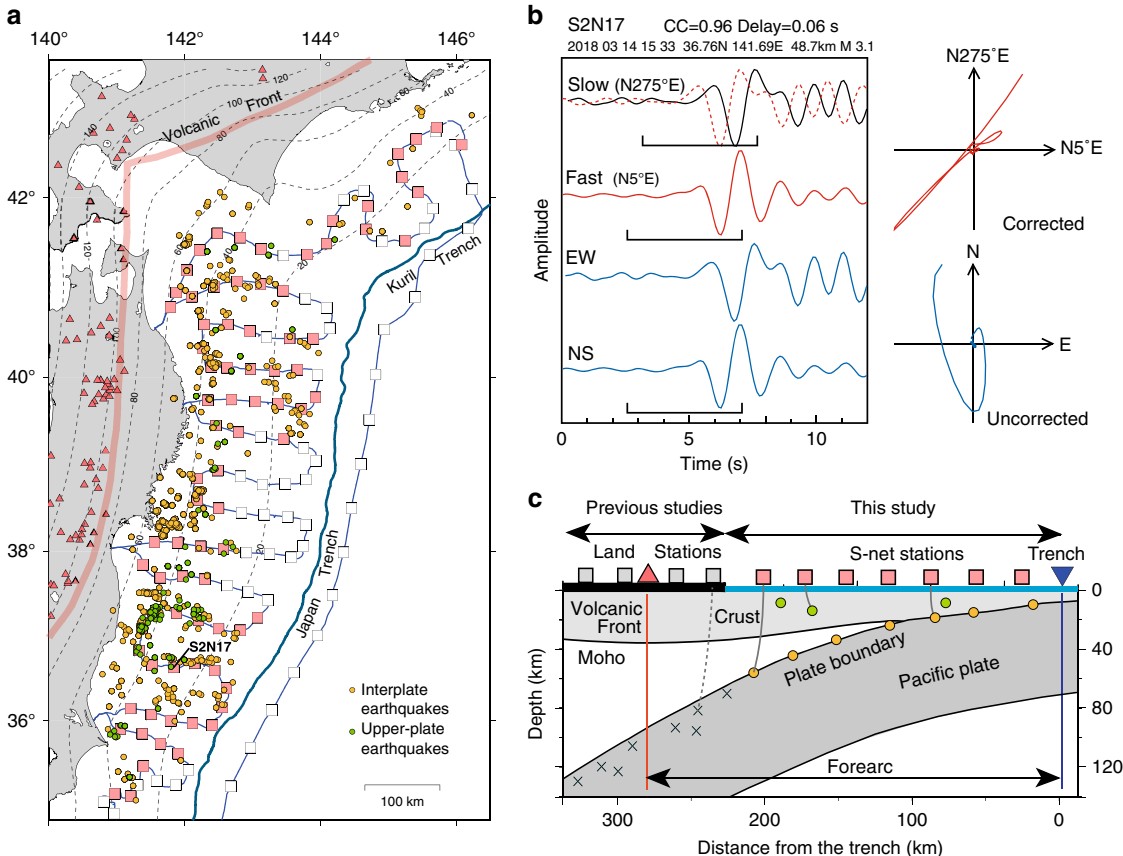

**Fig. 1 The S-net seafloor seismometers and the principle of shear-wave splitting analysis.** (**a**) Distribution of the S-net stations (squares) and the earthquakes used in this study. Orange and green circles show the interplate earthquakes and upper-plate earthquakes (depth ≤ 35 km), respectively. Red squares are the stations that have one or more splitting parameter estimations. Red triangles are the Quaternary volcanoes and thick pink lines show the volcanic front. Dashed contours show the depth to the subduction interface with an interval of 10 km. (**b**) Example of the procedure of the splitting parameter estimations. Left panel shows S waves. Bottom two traces show the original EW and NS components, and the top two traces show rotated waveforms to fast and slow directions. The dashed line in the slow direction is the waveform shifted with the amount of delay time. On the top, the station name, cross-correlation coefficient, delay time, and the earthquake source information are shown. The station location is also shown in (**a**). The right panel shows the particle motion for the original (uncorrected), and rotated and delay-time-corrected (corrected) waveforms. Note that the split shear-waves have similar waveforms in the fast- and slow- directions, and the cross-correlation value between the waveforms in the fast and slow directions becomes maximum for the direction and time shift. (**c**) Schematic cross-section view of the structure around 39°N [modified from Uchida et al., 2010[50]]. The source earthquakes are located on the subduction interface and the S-net stations are located above them. The S-net covers a large area in the forearc. Orange and green circles represent the offshore interplate and upper-plate earthquakes used in this study while crosses represent intraslab earthquakes used in previous studies.

of waveforms used by each station ranges from 1 to 101 (Supplementary Data 1 and 2), and station averages involving 10 or more waveforms are considered relatively reliable estimates. The splitting parameters at each station generally do not exhibit a dependence on incident angle and back-azimuth (Supplementary Fig. 2, Method).

The fast directions based on the interplate earthquakes have a spatially correlated pattern that tends to have NNE-SSW and ENE-WSW directions along the Japan and Kuril trenches, respectively (Fig. 2b), but the delay times are mostly around 0.1 s without significant spatial variations (Fig. 2b, length of bars). These directions are subparallel to the trench and the local strike of the subducting slab beneath the stations (Fig. 2a). The frequency distribution of the station-averaged fast directions offshore shows a clear peak in regions F1, F2, and F3 along the trench (Fig. 2b, black in the rose diagrams).

The used interplate events are located at ~10–50 km depths. Figure 3 shows the station-average splitting parameters as a function of the depth of the subduction interface (small red

circles). The source events are located within 45° from vertical beneath each OBS station, and the subduction interface depth approximately represents the length of the ray paths of these events. Note that the depth scale in Fig. 3b, c is not linear because it follows the shape of the subduction interface in Fig. 3a. Offshore, the fast directions relative to local slab strike at each station exhibit a concentration in the strike direction (i.e., around zero) (Fig. 3b) and delay times of 0.05–0.15 s. The average fast directions for every 20 km depth interval are stably within 10° of the slab strike in the offshore area (blue large circles at 0–40 km depth in Fig. 3b). The depth-averaged delay times for the offshore earthquakes are all ~0.1 s (blue large circles in 0–40 km depth in Fig. 3c). These results show that the offshore splitting parameters along the Japan and Kuril trenches are not sensitive to the depths of the earthquakes used.

The splitting parameters from the offshore upper-plate earthquakes averaged for each station also exhibit mostly trench-parallel fast directions (Fig. 2b, white bars). Although the number of available stations is small for these shallow earthquakes (N =

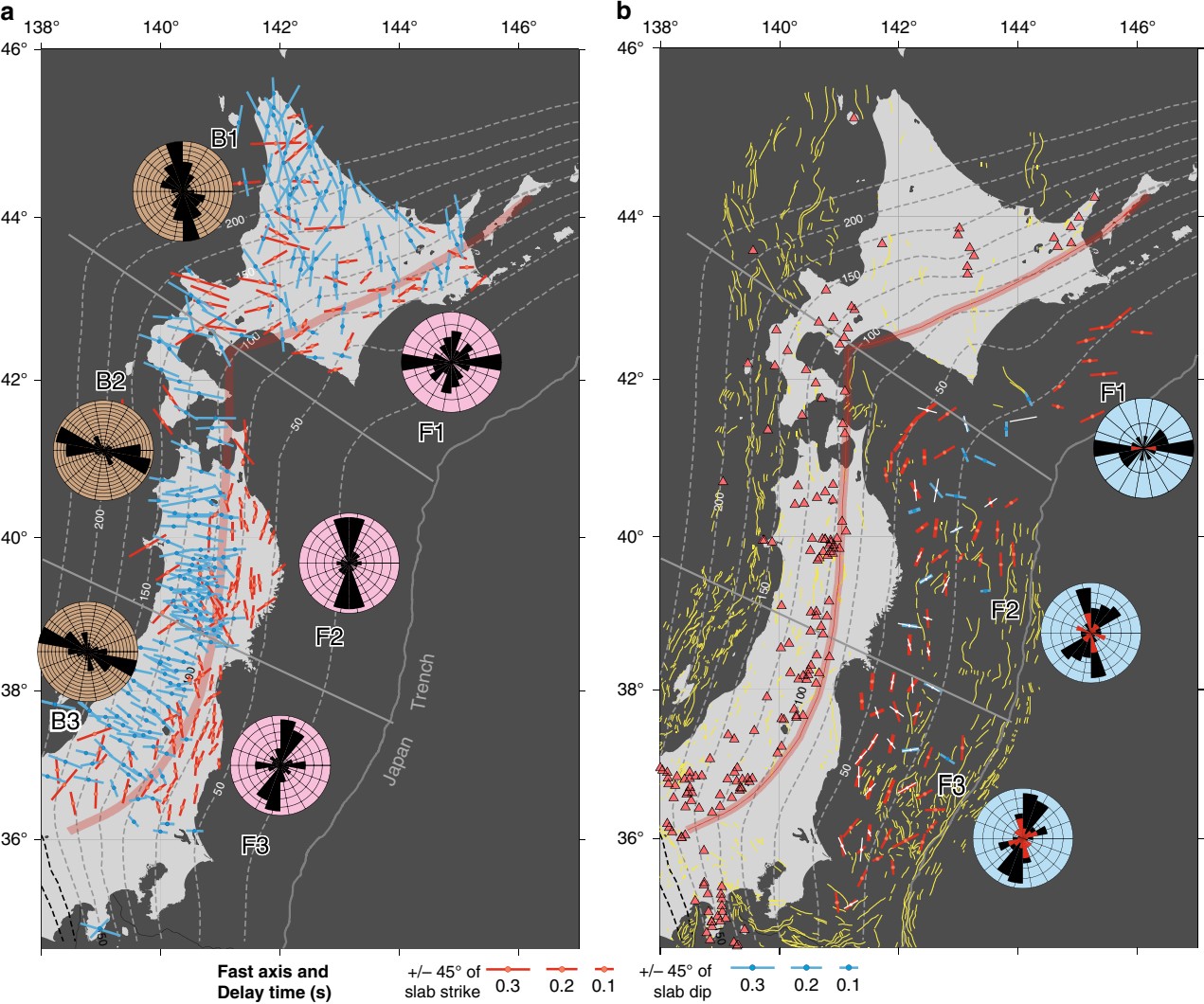

**Fig. 2 Distribution of the splitting parameters along the Japan and Kuril trenches.** (**a**) and (**b**) shows the fast direction (direction of the bars) and delay time (length of the bars) on land[17] and offshore, respectively. The averaged splitting parameters are plotted at each station. The color of bars indicate the direction is trench-parallel (red) or not (blue) for land intraplate and offshore interplate earthquakes. The results for offshore upper-plate earthquakes (depth ≤ 35 km) are shown in white. The thick bars offshore show the results estimated from ten or more earthquakes. The rose diagrams show the frequency distribution of the fast direction for the stations in areas F1–F3 and B1–B3 divided by the black lines, with blown, pink, and blue colors for the back-arc, onshore forearc, and offshore forearc areas, respectively. Black and red in the rose diagrams in (**b**) represent results for interplate and upper-plate earthquakes, respectively. The interval of the concentric circles in the rose diagrams is two. Black contours show the depth to the subduction interface in 25-km intervals[51,52]. Yellow lines show the active fault traces[45]. The volcanic front is marked by the thick pink line.

31), they clearly show fast directions (Fig. 2b) and delay times (Fig. 3b, c in green) similar to those of the deeper interplate sources.

Our study does not involve intraslab earthquakes and therefore excludes any contribution from the subducting slab (Fig. 3a). We can exclude significant shear-wave splitting in the forearc mantle wedge also for the following two reasons. (1) Our observed splitting parameters for the deeper interplate earthquakes and shallower upper-plate earthquakes in the offshore area are very similar (Fig. 3b, c). This suggests that the main source of anisotropy between the slab and surface is shallower than 35 km, primarily in the continental crust. (2) If the mantle-wedge part of the forearc had significant anisotropy, the delay time would increase with the depth of the subduction interface (Fig. 3a). But the observed delay times are depth-insensitive, and the splitting parameters are similar even in the area where the crust is very near or in direct contact with the slab (interface depth ≤ 35 km) (Fig. 3a). Therefore, the forearc trench-parallel fast directions

(schematically shown as double-headed arrows at the surface in Fig. 4) are explained by anisotropy in the overlying (upper-plate) crust (spheroids in Fig. 4). There is little shear-wave splitting within the forearc mantle wedge.

**Comparison with onshore forearc and back-arc.** Our offshore splitting parameters are consistent with those previously obtained onshore using the same method but with local intraslab earthquakes[16,17] (Fig. 2). The shear waves from intraslab earthquakes travel not only above but also within the subducting slab. However, their observed delay times do not depend on the length of the ray paths in the slab, and therefore the splitting in the slab is negligible if present at all [Fig. 4b of Nakajima and Hasegawa, 2004[16]]. Therefore, we can directly compare our results based on interplate earthquakes with those reported by Nakajima and Hasegawa [2004][16] and Nakajima et al. [2006][17] based on intraslab earthquakes.

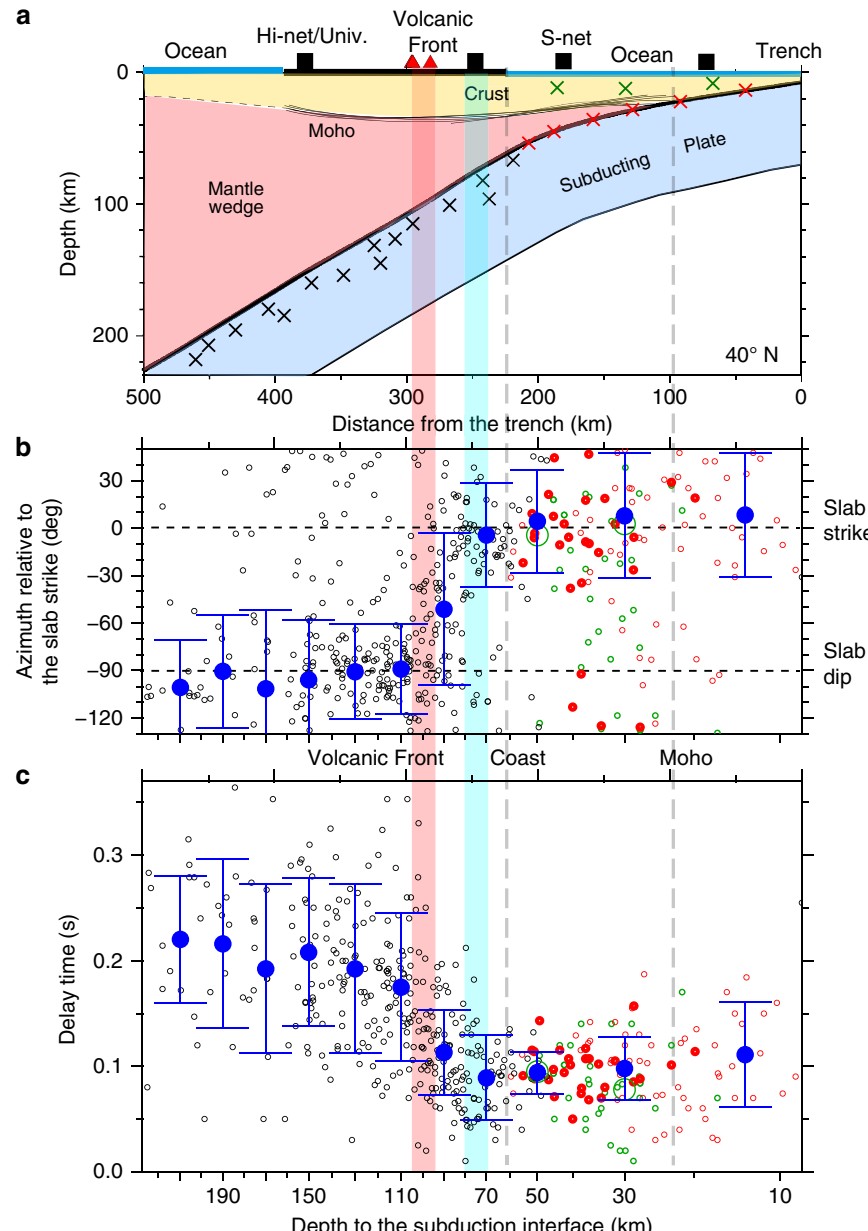

**Fig. 3 Dependence of splitting parameters on the depths of interplate earthquake sources.** (**a**) Geometry of subduction interface along a margin-normal profile at 40°N [Modified from Uchida et al., 2010[50]]. Crosses schematically show earthquakes, and black squares schematically show seismic stations that record shear waves from these events. (**b**) Azimuth of fast shear-wave polarization beneath each station measured clockwise from the strike direction of the Pacific slab. Note that an azimuth −130° is the same as 50°. (**c**) Delay time between the fast and slow polarized waves. The depth axis is scaled to distance using the interface geometry shown in (**a**). In (**b**) and (**c**), red, green, and black circles show station averaged values based on interplate events, upper-plate events, and intraslab events, respectively. The intraslab data are from Nakajima et al. [2006][17]. Filled red circles indicate the number of waveforms is larger than 10. Blue circles with error bars (one standard deviation) represent average values for events within every 20 km depth interval, except for the shallowest one which is an average from the 7 km trench depth to 20 km. For each depth average, we show the mean azimuth and circular standard deviation that utilize the azimuth and length of the summed unit vectors[53]. Blue and red vertical bars mark the maximum depth of slab-mantle decoupling[33] and the volcanic front, respectively.

The fast directions are predominantly trench-parallel in both the onshore and offshore parts of the forearc (Fig. 2, to the east of the thick pink line). The frequency distribution of the fast directions offshore (the rose diagrams with blue background in Fig. 2b) and those for the narrow land area of the forearc (the rose diagrams with pink background in Fig. 2a) have similar peak azimuth of the directions. This azimuthal preference can also be seen from the fast direction relative to the slab strike plotted as a function of the subduction interface depth (Fig. 3b; there is no

systematic difference between the red and offshore black circles). Not only the azimuths of the fast directions but also the delay times (Fig. 2, length of bars) show similar values between the offshore and onshore parts of the forearc. The delay times plotted against the depth of the subduction interface (Fig. 3c) show little change in the depth range of 0–80 km, including both the land and offshore stations.

The volcanic front, the boundary dividing the forearc and back-arc areas, is located above where the slab is at depths of

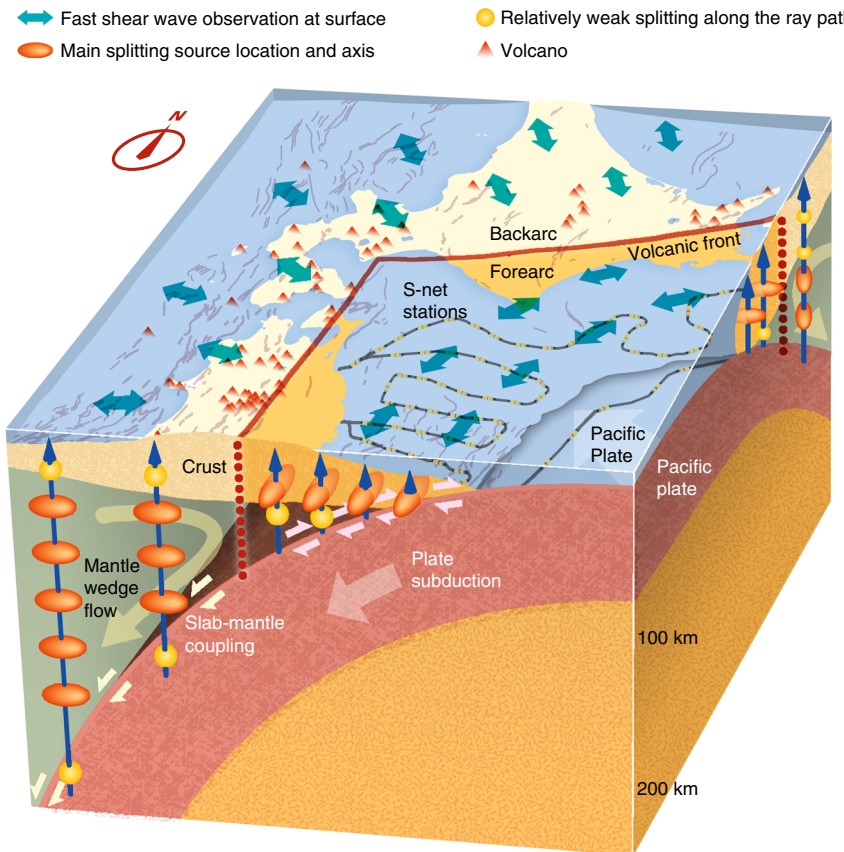

**Fig. 4 Schematic illustration of shear-wave anisotropy in the Japan Trench subduction zone.** Green double-headed arrows show the observation of the fast shear wave direction at the surface stations. Spheroids show the main location of the anisotropy along ray paths (blue arrows), with the longest axis indicating the fast direction. Spheres represent a lack of anisotropy. Grayline and cylinders show the S-net cable and OBSs, respectively. Redline and red dashed lines show the volcanic front and its deep extension. Pairs of opposite pink arrows show a slip of the subduction interface, but pairs of arrows with the same direction show the mantle material is coupled to, and travels with, the slab (interplate coupling). Triangles and thin gray lines represent volcanoes and active faults, respectively.

90–100 km (Figs. 1a and 2). The splitting pattern in the back-arc is entirely different from the forearc, with the fast directions being trench-normal and the delay time being ~0.2 s (Figs. 2 and 3).

## Discussion

The most important finding of this work is the lack of significant shear-wave splitting in the forearc mantle wedge which indicates a lack of anisotropy. Anisotropy with a sub-vertical symmetry axis may also be invoked to explain the lack of splitting, but the consistency of this pattern over the entire ~700 km of the margin including the kink between the Kuril and Japan trenches makes this possibility extremely unlikely.

Our new results together with earlier results based on land stations thus identify a sharp anisotropy contrast between the forearc and back-arc parts of the mantle wedge (Figs. 2 and 4). The back-arc features trench-normal fast direction (Fig. 2a) with the source of the anisotropy residing in the mantle wedge (spheroids in Fig. 4)[17]. The trench-normal fast direction is most logically explained as reflecting lattice-preferred orientation of olivine minerals caused by slab-driven mantle wedge flow (Fig. 4, large arrows). In the forearc, however, the S-net results suggest no or very weak anisotropy in the mantle wedge.

In a previous study[17], limited to land areas, the forearc anisotropy was speculated to be due to the presence of B-type olivine in the mantle wedge. It was assumed that the forearc mantle wedge was also involved in the slab-driven wedge flow which would produce the same lattice-preferred orientation of olivine as in the back-arc mantle wedge. In this situation, abundant presence of the B-type olivine, expected for a high-water content and high-stress condition, would produce a fast direction perpendicular to the flow direction[32]. Since the forearc mantle wedge is known to be cold and thus unlikely to engage in vigorous thermally activated creep, Kneller et al. [2005][14] assumed that it deformed very slowly due to coupling with the slab to allow olivine minerals to be oriented in the trench-normal direction, so that the B-type olivine could still produce trench-parallel fast directions.

In contrast, Wada and Wang (2009)[33] and Wada et al. (2011)[34] inferred from heat flow and other geophysical observations that the slab and the mantle are fully decoupled until a depth of 70–80 km (pink arrows along plate interface in Fig. 4) but fully coupled at greater depths (yellowish arrow pairs in Fig. 4), a notion that is supported by other studies[35–39]. Consequently, the back-arc mantle wedge is expected to engage in full-speed viscous wedge flow producing trench-normal fast directions, but the forearc mantle wedge is expected to be fully stagnant producing no flow-related anisotropy. The lack of anisotropy in the stagnant forearc mantle wedge in contrast with the presence of strong strike-normal anisotropy in the flowing back-arc is not only observed in NE Japan where the subducting slab is very old and cold but also consistent with observations from Cascadia which is an end-member warm-slab subduction zone[5,40]. Therefore, the mantle wedge dynamics inferred from the mapping of mantle wedge

anisotropy in NE Japan is likely ubiquitous for subduction zones regardless of their thermal state.

The origin of the offshore crustal anisotropy delineated in this study is of secondary significance to the mantle-wedge focus of this paper, although it is important in the study of crustal dynamics. The predominantly trench-parallel direction is similar to previous estimates beneath the land area using crustal earthquakes[15,41,42] and a 3D model of azimuthal anisotropy tomography[43]. Crustal anisotropy is often explained by stress-controlled crack alignment or by structural fabrics[6,41,44]. Figure 2b shows that the fast directions determined in this work are consistent with the prevalence of trench-parallel-striking active crustal faults[45]. Exceptions near the boundary of regions F1 and F2 are compatible with the presence of NNW-SSE trending trust faults due to arc-arc collision in this corner by the sliver motion of the Kuril forearc[46]. The comparison supports the notion that the structural fabrics are responsible for the overall trench-parallel crustal anisotropy. Nevertheless, some contribution from horizontal stress cannot be fully excluded.

## Methods

**The selection of the source earthquakes.** We used interplate and upper-plate earthquakes to determine the location of the main splitting. For the interplate earthquakes, we used earthquakes with interplate type focal mechanism and those identified as repeating earthquakes. The focal mechanisms are provided by the Full Range Seismograph Network of Japan (F-net)[26]. To select interplate earthquakes we employ the criteria used by Asano et al. [2011][47] and Hasegawa et al. [2012][48]: rake angle of > 0°, and three-dimensional (3-D) rotation angle[49] of the focal mechanism relative to that of the reference interplate earthquake of < 35°; and depth separation of the centroid from the plate interface of < 20 km. The repeating earthquakes are selected based on waveform similarity at land stations and represent an update of the catalog of Uchida and Matsuzawa [2013][28]. We use 40 s window to calculate waveform coherence, and select repeater pairs if the coherence is 0.8 or larger in the frequency range around their corner frequencies[28]. For the earthquakes in the upper plate, we also used earthquakes that have focal mechanisms by F-net. They have selected if the focal depths are shallower than the subduction interface or 35 km, whichever is shallower, and if the focal mechanisms are not the interplate type according to the criteria described above.

**Waveform rotation.** The original S-net data provided by National Research Institute for Earth Science and Disaster Resilience are velocity waveforms for the X, Y, and Z axes. The X-axis is along the long axis of the cable and Y and Z axes are perpendicular to that direction. Since we need horizontal components to perform shear-wave splitting analysis, we rotated the waveforms to East, North, and Up (ENU) directions. The rotation matrix which is estimated from the gravity and teleseismic long-period Rayleigh waves observed by accelerometer[24] was used for the conversion from the XYZ to ENU components. The accuracy of the azimuth of the seismometers are estimated to be 3–12°[24]. Takagi et al.[24] also found that some large earthquakes resulted in the rotation of the observation pressure vessel. The $M_W$ 6.0 off-Sanriku earthquake on August 20, 2016 and the $M_W$ 6.9 off-Fukushima earthquake on November 22, 2016, caused rotation with 1 degree or more for 1 station and 3 stations, respectively. Considering these changes, we used daily estimates of the rotation matrix when converting the waveforms.

**Estimation of the splitting parameters.** The estimation of the fast directions and delay times is performed using a cross-correlation method[30] that takes advantage of the similarity of fast and slow shear waves when the waveforms are rotated to the vibration direction of the fast and slow shear waves. We used the bandpass filtering of 2–8 Hz which is the same as used in Nakajima et al. [2006][17]. The waveforms are rotated in 5° steps in the 0 to 175° range, and the rotated waveforms are shifted in 0.01-s steps in the 0 to 1 s range, to find the largest cross-correlation value. The amount of time shift and rotation angle when the cross-correlation value is largest are regarded as the delay time and the fast direction, respectively. The time window is 1–1.5 cycles of the waveform oscillations. We visually checked the seismograms one by one carefully and only used those that showed clear enough S arrival to be used for the shear-wave splitting analysis. Figure 1b shows an example of the procedure as described in the main text. The horizontal particle motion of the original waveforms exhibits an elliptical shape, and the waveform after the removal of the effect of anisotropy is almost linear (Fig. 1b). Typical uncertainties for individual fast direction and delay time estimated by the t-test are less than 30° and 0.03 s, respectively. The fast directions and delay times for each station show a standard deviation of 33° and 0.06 s, respectively, on average for the interplate events (Supplementary Data 1). The upper-plate events show similar values on average (31° and 0.06 s for the fast direction and delay time, respectively, Supplementary Data 2).

## Data availability
The S-net data are available at https://hinetwww11.bosai.go.jp/auth/ subject to the policies of National Research Institute for Earth Science and Disaster Prevention (NIED). The focal mechanism catalog used in this study is available at http://www.fnet.bosai.go.jp/ subject to the policies of the NIED. The splitting data generated and analyzed during this study are included in the Supplementary Data.

## Code availability
The analysis codes used in this study are available from the corresponding author upon request.

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

## Acknowledgments

This work was supported in part by JSPS KAKENHI 15K05260, 16H06473, 17KK0081, and 19H05596 and MEXT of Japan, under its Earthquake and Volcano Hazards Observation and Research Program. We thank Ryosuke Azuma, Akira Hasegawa, Toru Matsuzawa, Martha Savage, Katsuhiko Shiomi, and Genti Toyokuni for their fruitful discussions and comments. We also thank Kaoru Sawazaki for the technical information on the S-net seismometers, Sachiko Tanaka for discussion and comments on the manuscript and Atsushi Otomo at Design Convivia for the skillful illustration of Fig. 4.

## Author contributions

N.U. analyzed the waveform data, selected repeating earthquakes, and estimated the splitting parameters. N.U. wrote the manuscript with K.W. N.U, J.N., and Y.A. decided the direction of analysis and cooperated in the analysis. N.U., K.W., R.T., and R.H. contributed to the improvements of the figures. R.T. and T.N. helped the conversion of the waveform data and made event data. K.Y. selected the interplate and overriding plate earthquakes analyzed in this study and provided information on crustal stress. R.H. and T.O. directed this project. Y.A. helped to prepare the continuous waveform and checked its quality. All authors discussed the results and commented on the manuscript.

## Competing interests

The authors declare no competing interests.
