## [Peer Review File · Nature Communications]

Reviewers' comments:

Reviewer #1 (Remarks to the Author):

Summary:

Uchida et al. present interesting and important new results of seismic anisotropy derived from shear-wave splitting measurements in the forearc region of NE Japan. The findings are based on a unique data set that includes recordings from local interplate earthquakes at the recently installed S-net, a cabled seafloor seismic network, consisting of 150 stations covering a subsea area of about 300 × 1000 km. The dataset further includes shallow earthquakes in the overriding plate which allow for a more precise depth estimation as to the source of the anisotropy.

The shear-wave splitting measurements obtained at the OB seismic stations are complemented by results from land stations (in both the forearc and backarc regions) obtained and presented in several earlier studies such that the interpretation of the observations is based on a comprehensive data set.

The splitting parameters in the forearc and backarc regions of the Japan and Kuril trenches are distinctly different: The fast-polarization orientations are predominantly trench-parallel in both the onshore and offshore parts of the forearc with delay times of about 0.1 sec, whereas the backarc exhibits trench-perpendicular orientations and larger delay times (~ 0.2 sec).

In the forearc, the authors exclude a significant contribution from the mantle wedge, based on splitting parameters (delay times) that do not depend on earthquake depth. Therefore, the corresponding anisotropy is assigned to the crust. In the backarc, however, the trench-perpendicular anisotropy is explained by slab-driven flow in the mantle wedge.

Comments:

The observations and conclusions presented here are certainly of great importance and are of interest to a wide audience. The figures are generally instructive. Extending the analysis of anisotropy to the offshore region of the forearc provides a unique opportunity.

However, I do have some doubts regarding the claims of the authors that the results will "unambiguously resolve the controversy regarding the mechanism of forearc anisotropy in this region" and "Given the consistency of our results with generic geodynamic models of subduction zones, the knowledge learned is expected to be globally applicable" (from page 5).

First, much of the controversy involving the interpretation of shear wave splitting measurements in subduction zones is based on analyses of local as well as teleseismic observations. The analysis of local data, alone, cannot resolve the possible role of mantle flow beneath the slab, as first hypothesized by Russo & Silver for South America. While the controversial evidence for anisotropy in subduction zones is pointed out and described in the introduction to paper, the authors fail to make use of the full potential of their seismic network. It would certainly be helpful to also include measurements from teleseismic phases, such as SKS. Delay times obtained from teleseismic shear-wave splitting above subduction zone are, typically, in the order of 1 sec or larger and, in view of the much smaller delay times reported here, the question remains where the main source of the anisotropy (accounting for > 80% of the delay!) resides. Also, global applicability of the findings may be possible but still needs to be shown.

In my view, the presentation could be improved by concentrating on the, certainly significant, findings and conclusions that can be drawn from the current observations. In the present form, the introduction to the manuscript and the findings described do not quite match.

While I believe that the data analysis is generally sound, I wonder about the example presented in Figure 1b. The blue lines depict the original (uncorrected) horizontal shear phases. In contrast to the findings, it

seems that the EW component arrives early (and the NS component is delayed). Waveforms of the corrected phases (in fast-slow coordinates) are very similar, which points to the fact that the fast and slow directions are relatively close to the geographic axes. However, in the panel above, the fast direction is indicated as N5E, which differs from NS by only 5 degrees (in contrast to the impression given by the "naked eye" that EW is fast). I assume that labels EW and NS should be exchanged? Also, to clarify the time distance between ticks shown, a more conventional label for the time axis should be used. The caption should indicate where (geographically) the station is located.

The significance of the "repeating earthquakes" for the anisotropy results presented here does not become clear.

Minor:

Page 2 "world's first" - > first

Page 3: "The fast direction from teleseismic events is also trench-parallel in many subduction zones". This statement seems a bit outdated. There are now other examples that show a wider variety and trench perpendicular fast directions.

Page 4: mid-ocean ridge

Page 4: "The remarkable non-uniqueness of the interpretations seen in the literature is to a large part due to limitations in the spatial coverage and reliability of the observations."

This is a very general statement and I would not judge the "reliability of the observations" reported in the previous studies in this manner.

Page 11: "azimuthal anisotropy tomography" -> anisotropic surface-wave tomography (?)

Caption Figure 1: "The dashed line" -> Dashed contour

Reviewer #2 (Remarks to the Author):

Uchida et al presented one of the first systematic mappings of offshore fore-arc anisotropy using shear wave splitting measurements from local earthquakes (interplate and upper-plate) recorded by offshore S-net ocean bottom cable network. Their result indicates clear contrast between the stagnant fore-arc wedge corner (little anisotropy) and the rest of the mantle wedge (strong anisotropy), which forms the major finding of the study.

major comments:

The technique of shear wave splitting has been widely used to constrain seismic anisotropy on earth. To properly conduct splitting measurements using S-net, the authors have previously conducted systematic analysis to examine sensor mis-orientation, which allows them to conduct meaningful and accurate measurement.

Typically, their sea-floor splitting observations indicate trench-parallel fast splitting direction, which is different from the trench-normal fast splitting direction measured against land stations in the arcback arc region. As highlighted earlier, the writing seems to deserve some improvement, mostly on the focus of discussion (see below) and a more broader range of discussions.

0. While offshore S-net splitting measurements are extremely valuable in mapping seismic anisotropy and understanding their underlying cause, the result will be much more impactful if the authors can compare their result against other area equipped with splitting measurements on the seafloor. One example is Cascadia subduction zone, where OBS network, the Cascadia initiative, and the USArray have produced invaluable splitting measurements as well. In particular, contrasting results near young subducting slab

(Juan-de-Fuca plate) against those in old subducting slab (e.g., Pacific) potentially allow a more robust discussion contrasting anisotropy in the fore-arc wedge corner and the rest of the mantle wedge.

1. The authors average splitting parameters of the earthquakes of the same type. Are these earthquakes located at similar back-azimuth? Is there any back-azimuthal dependence in splitting parameters? Presumably, such info should help dissect anisotropy symmetry and discuss the cause of anisotropy.
2. The depth of earthquake is relevant to the inferred regions of sampling. With high quality P and S arrivals recorded by the S-net, can the authors verify (in some cases) that the focal depth reported by F-net is largely consistent with observed P-S time?
3. While it is clear that the splitting time is relatively small near the stagnant wedge corner, it does not always reflect the strength of anisotropy. Another likely scenario is that the symmetry axis is sub-vertical, given the thin wedge corner, it can result in apparent isotropy. As serpentinised mantle wedge is often invoked to interpret a hydrate wedge corner and decoupling, it is probably relevant to discuss the observations with respect to the anisotropy symmetry of serpentine.

minor comments:

1. line 54-60, the authors illustrate several source areas that may contribute to splitting observations in the fore-arc area. However, the mechanism producing seismic anisotropy discussed in each area is quite limited. Several alternative mechanisms that may account for the observations deserve to be noted. A broader discussion here will also help the authors to examine their result in the context of alternative scenarios. On the other hand, it is not clear if the new observations, while very valuable, can necessarily distinguish alternative mechanisms for a given area. To clarify, the authors identified likely area resulting in observed splitting, but they did not attempt to use their measurement to distinguish alternative mechanism in a given area.
2. line 54-60, and line 64-71. These two sections may deserve some rearrangement. Given the local splitting measurement from interpolate and upper plate events, it seems that sub-slab and slab anisotropy do not come into play in this work since they are not sampled. It is unclear the writing is consistent with the focus of this work. Similarly, discussions in line 189-191 seem not relevant.
3. White dot shown in Fig. 2b can be a bit misleading (e.g., null measurement with zero splitting time). Since there is no upper plate earthquake close to the trench, it is probably better to simply display inter-plate earthquake measurement (no white bar, or white dot).
4. It will be useful to display the distribution of incident angle at several representative stations.
5. In Fig. 3b, fast splitting directions around ~110-170 km from the trench vary more significantly than those in other fore-arc area, but the error bar does not seem to reflect this?

Reviewer #3 (Remarks to the Author):

The manuscript "Subduction dynamics from forearc anisotropy mapped with S-net seafloor seismometers" presents evidence for widespread trench parallel fast axes in the crustal forearc of the Japan subduction zone from the systematic analysis of a globally relevant and novel dataset. The authors also make a strong case for limited anisotropy in the stagnant mantle wedge. I believe that this paper will find widespread interest in seismotectonics, geodynamics and geologic communities and I support its publication with minimal revision. This work will also go a long way to constraining geodynamic models of Japanese crustal deformation -- necessary to understanding the seismic hazard consequences of the 2011 great Tohoku earthquake.

The paper is well-written. The language is clear and succinct. My minor grammatical suggestions are tracked in the manuscript. The figures are high quality and easy to understand.

This work is notable and to be credited for its simplicity of observations. The basis for the paper is almost a direct observation. Previous criticisms of isolating layer anisotropy are both its path-integrated nature and the regularization necessary to perform anisotropy tomography. The current project overcomes these issues by brute force and abundance of data.

The two key conclusions, and each individually would warrant publication, are 1) persistent trench parallel fast axes in the upper plate and 2) lack of strong anisotropy in the stagnant mantle wedge.

Of note, there have previously been other consistent observations for strike-parallel fast axes in the forearc crust, but this is by far the most compelling, owing mostly to the data available for analysis. In fact, Fry et al., (2010) published direct evidence of inherited trench parallel fast directions for European subduction (doi:10.1016/j.epsl.2010.06.008).

Please see appended additional information for a comparison of those results.

When ascribing the source of crustal anisotropy, please describe "structural fabrics". Are these embricate structural domains? Pervasive and oriented patterns of crustal deformation? More and more evidence is suggesting that fluids escaping from the slab may travel in trench parallel deformation bands. Perhaps this is a direction for further enquiry with the S-Net data.

Have the authors thought about the competing trench sub-perpendicular fast axes in domains F1 and F3? I would like to see at least mention of these trends.

On page 4, ". The remarkable non-uniqueness of the interpretations seen in the literature is to a large part due to limitations in the spatial coverage and reliability of the observations.",
A bigger part of the problem is the limited vertical resolution (not spatial coverage) of splitting techniques – observations can be viewed as a path-integrated average.

In all cases, I would like to see "fast direction" replaced with "fast axis", as reciprocity means anisotropy is not a vector. The use of "direction" in the literature is widespread and the authors can continue to do so, but my suggestion is that "axis" is more appropriate.

Reply to the reviewer 1

Thank you for the review. Below I show your comments in orange and our replies in black. The line number is based on the annotated pdf file.

Comment 1:

Summary:

Uchida et al. present interesting and important new results of seismic anisotropy derived from shear-wave splitting measurements in the forearc region of NE Japan. The findings are based on a unique data set that includes recordings from local interplate earthquakes at the recently installed S-net, a cabled seafloor seismic network, consisting of 150 stations covering a subsea area of about 300×1000 km. The dataset further includes shallow earthquakes in the overriding plate which allow for a more precise depth estimation as to the source of the anisotropy.

The shear-wave splitting measurements obtained at the OB seismic stations are complemented by results from land stations (in both the forearc and backarc regions) obtained and presented in several earlier studies such that the interpretation of the observations is based on a comprehensive data set.

The splitting parameters in the forearc and backarc regions of the Japan and Kuril trenches are distinctly different: The fast-polarization orientations are predominantly trench-parallel in both the onshore and offshore parts of the forearc with delay times of about 0.1 sec, whereas the backarc exhibits trench-perpendicular orientations and larger delay times (~ 0.2 sec).

In the forearc, the authors exclude a significant contribution from the mantle wedge, based on splitting parameters (delay times) that do not depend on earthquake depth. Therefore, the corresponding anisotropy is assigned to the crust. In the backarc, however, the trench-perpendicular anisotropy is explained by slab-driven flow in the mantle wedge.

Reply:

Thank you for understanding the results.

Comment 2:

The observations and conclusions presented here are certainly of great importance and are of

interest to a wide audience. The figures are generally instructive. Extending the analysis of anisotropy to the offshore region of the forearc provides a unique opportunity.

However, I do have some doubts regarding the claims of the authors that the results will “unambiguously resolve the controversy regarding the mechanism of forearc anisotropy in this region” and “Given the consistency of our results with generic geodynamic models of subduction zones, the knowledge learned is expected to be globally applicable” (from page 5).

First, much of the controversy involving the interpretation of shear wave splitting measurements in subduction zones is based on analyses of local as well as teleseismic observations. The analysis of local data, alone, cannot resolve the possible role of mantle flow beneath the slab, as first hypothesized by Russo & Silver for South America. While the controversial evidence for anisotropy in subduction zones is pointed out and described in the introduction to paper, the authors fail to make use of the full potential of their seismic network. It would certainly be helpful to also include measurements from teleseismic phases, such as SKS. Delay times obtained from teleseismic shear-wave splitting above subduction zone are, typically, in the order of 1 sec or larger and, in view of the much smaller delay times reported here, the question remains where the main source of the anisotropy (accounting for > 80% of the delay!) resides. Also, global applicability of the findings may be possible but still needs to be shown.

In my view, the presentation could be improved by concentrating on the, certainly significant, findings and conclusions that can be drawn from the current observations. In the present form, the introduction to the manuscript and the findings described do not quite match.

Reply: Much of the concerns here are caused by our failure to explain the objective of the paper more clearly. In particular, as correctly pointed out at the end of the comment above, we did a poor job describing the main focus of the paper in the Introduction part. Our focus is exclusively on the mantle wedge with no intention to resolve the entire problem of forearc anisotropy involving the slab and sub-slab mantle flow. Therefore, teleseismic data are not relevant to this work. We have now modified numerous parts of the paper, even including a change in the title of the paper, to clarify the mantle wedge focus. Some but not all of the numerous changes for this reason are highlighted below.

Title:

Mantle wedge dynamics from forearc anisotropy mapped with S-net seafloor seismometers

Abstract:

Knowledge of shear-wave anisotropy is important to understanding the structure and dynamics of subduction zone mantle wedge.

Lines 68-73 (introductory paragraph):

In understanding subduction zone dynamics, it is important to know the pattern of solid flow within the mantle wedge between the upper plate and subducting slab, and seismic anisotropy is an excellent indicator of the flow pattern. At present, the anisotropy of the offshore part of the forearc mantle wedge is essentially unknown because of lack of seafloor seismic observations.

Lines 79-90: *From these results, it is almost impossible to infer the anisotropy state of the forearc mantle wedge with confidence. Many studies assume the source of the observed anisotropy to be outside of the mantle wedge, either beneath the subducting slab associated with trench-parallel mantle flow¹⁰, within the slab associated with structural fabrics acquired before subduction¹¹ or upon subduction due to plate bending⁹, or within the overlying crust associated with stress-controlled preferred orientation of microcracks or geological fabric^{6,12,13}. Those that consider the anisotropy to be within the forearc mantle wedge often associate it with assumed abundance of B-type olivine minerals that are aligned by the mantle wedge corner flow¹⁴. The remarkably poor knowledge of mantle wedge anisotropy is to a large part due to limitations in the spatial coverage of the observations and the vertical resolution of the anisotropy analysis.*

Lines 96-98: *It is also unresolved whether the source of the trench-parallel fast axis is uniform in depth beneath the forearc.*

Comment 3:

While I believe that the data analysis is generally sound, I wonder about the example presented in Figure 1b. The blue lines depict the original (uncorrected) horizontal shear phases. In contrast to the findings, it seems that the EW component arrives early (and the NS component is delayed). Waveforms of the corrected phases (in fast-slow coordinates) are very similar, which points to the fact that the fast and slow directions are relatively close to the geographic axes. However, in the panel above, the fast direction is indicated as N5E, which differs from NS by only 5 degrees (in contrast to the impression given by the “naked eye” that EW is fast). I assume that labels EW and NS should be exchanged? Also, to clarify the time distance between ticks shown, a more conventional label for the time axis should be used. The caption should indicate where (geographically) the station is located.

Reply:

Thank you for finding this mistake. In the submitted version, the EW and NS labels was mistakenly exchanged. We have now corrected it. We have also changed the label for the time axis to be more conventional. We have added the station name in Fig.1a and explained it in the caption.

Comment 4:

The significance of the “repeating earthquakes” for the anisotropy results presented here does not become clear.

Reply:

The clarification of the mantle wedge focus has automatically explained why the repeating earthquakes are used. To further clarify this, we have added the following sentence in lines 313-315.

Lines 314-316:

In addition, we also use the 321 small repeating earthquakes ($M \geq 2.5$) that have been identified to be located along the creeping parts of the subduction interface²⁸ (Fig. 1a) to greatly increase the number of available interplate earthquakes.

Comment 5:

Minor:

Page 2 “world’s first” - > first

Reply:

We have changed the wording as suggested.

Comment 6:

Page 3: “The fast direction from teleseismic events is also trench-parallel in many subduction zones”.

This statement seems a bit outdated. There are now other examples that show a wider variety and trench perpendicular fast directions.

Reply: Thank you for this suggestion. We have revised the text as follows.

Lines 78-79

“The fast axes from teleseismic events are also trench-parallel in some subduction zones⁹.”

Comment 7:

Page 4: mid-ocean ridge

Reply: Thank you for finding this typo. We have removed this part due to the re-organization of the introduction.

Comment 8:

Page 4: “The remarkable non-uniqueness of the interpretations seen in the literature is to a large part due to limitations in the spatial coverage and reliability of the observations.”

This is a very general statement and I would not judge the “reliability of the observations” reported in the previous studies in this manner.

Reply: According to your comment and a comment from Reviewer #3, we have rephrased the sentence as follows.

Lines 87-90 “*The remarkably poor knowledge of mantle wedge anisotropy is to a large part due to limitations in the spatial coverage of the observations and the vertical resolution of the anisotropy analysis.”*

Comment 9:

Page 11: “azimuthal anisotropy tomography” → anisotropic surface-wave tomography (?)

Reply: With a better clarified focus, we find this paragraph redundant and thus have removed it.

Comment 10:

Caption Figure 1: “The dashed line” → Dashed contours

Reply: We have changed the words as suggested.

Reply to the reviewer 2

Thank you for the review. Below I show your comments in orange and our replies in black. The line number is based on the annotated pdf file.

Comment 1:

Uchida et al presented one of the first systematic mappings of offshore fore-arc anisotropy using shear wave splitting measurements from local earthquakes (interplate and upper-plate) recorded by offshore S-net ocean bottom cable network. Their result indicates clear contrast between the stagnant fore-arc wedge corner (little anisotropy) and the rest of the mantle wedge (strong anisotropy), which forms the major finding of the study.

Reply:

Thank you for your comments.

Comment 2:

major comments:

The technique of shear wave splitting has been widely used to constrain seismic anisotropy on earth. To properly conduct splitting measurements using S-net, the authors have previously conducted systematic analysis to examine sensor mis-orientation, which allows them to conduct meaningful and accurate measurement.

Typically, their sea-floor splitting observations indicate trench-parallel fast splitting direction, which is different from the trench-normal fast splitting direction measured against land stations in the arc-back arc region. As highlighted earlier, the writing seems to deserve some improvement, mostly on the focus of discussion (see below) and a more broader range of discussions.

Reply:

Thank you for this suggestion. We will reply individual comments below.

Comment 3:

0. While offshore S-net splitting measurements are extremely valuable in mapping seismic anisotropy and understanding their underlying cause, the result will be much more impactful if the authors can compare their result against other area equipped with splitting measurements on the seafloor. One example is Cascadia subduction zone, where OBS network, the Cascadia

initiative, and the USArray have produced invaluable splitting measurements as well. In particular, contrasting results near young subducting slab (Juan-de-Fuca plate) against those in old subducting slab (e.g., Pacific) potentially allow a more robust discussion contrasting anisotropy in the fore-arc wedge corner and the rest of the mantle wedge.

Reply: Thank you for this suggestion. For Cascadia, previous work based on limited land and OBS data also suggest margin-parallel and margin-normal fast axes for the forearc crust and back arc mantle, respectively. We have added the sentence below near the end of the main text to reference these previous studies. More recent OBS studies at Cascadia, such as Bodmer et al. (Geology 2015), VanderBeek and Toomey (GRL, 2017; JGR, 2019), are not directly related to the study of the mantle wedge, and are therefore not cited.

Line 540-546: “The lack of anisotropy in the stagnant forearc mantle wedge in contrast with the presence of strong strike-normal anisotropy in the flowing back arc are not only observed in NE Japan where the subducting slab is very old and cold but also consistent with observations from Cascadia which is an end-member warm-slab subduction zone^{5,45}. Therefore, the mantle wedge dynamics inferred from the mapping of mantle wedge anisotropy in NE Japan is likely ubiquitous for subduction zones regardless of their thermal states.”

Comment 4:

1. The authors average splitting parameters of the earthquakes of the same type. Are these earthquakes located at similar back-azimuth? Is there any back-azimuthal dependence in splitting parameters? Presumably, such info should help dissect anisotropy symmetry and discuss the cause of anisotropy.

Reply: We have made a new figure showing the splitting parameters with back-azimuth for 16 stations. We could not find any clear systematic back-azimuthal dependence in splitting parameters. We have added the figure as Fig. S1 and a comment as follows.

Lines 348-420 “The splitting parameters at each station generally do not exhibit a dependence on incident angle and back-azimuth (Supplementary Figure 2, Method).”

Comment 5:

2. The depth of earthquake is relevant to the inferred regions of sampling. With high quality P and S arrivals recorded by the S-net, can the authors verify (in some cases) that the focal depth reported by F-net is largely consistent with observed P-S time?

Reply: It is true that the depth of earthquake is relevant to inferred region of sampling. For the interplate events, however the depth information is not very important because the depth is only used for excluding events that are 20km or more distant from the plate interface and the interplate type focal mechanism is the main selection criteria. Please also note that we did not use focal depth in any figure but only used the depth to the plate boundary in Fig. 2. For shallow earthquakes (depth ≤ 35 km, above the plate boundary and not having interplate type focal mechanism), the depth information is relatively important for distinguishing between intraslab and interplate earthquakes. Following your suggestion, we checked the F-net depth by reading the S-P time for 19 earthquakes below the S-net stations (incident angle less than 5 degree). The results show that the F-net depths for interplate and shallow earthquakes are consistent with the S-net S-P time considering the near surface sediments with low S-wave velocity. It is considered that the F-net data have better depths than JMA, because F-net's CMT inversion fits the 'depth phase' in the waveform. A new figure showing this result has been added as a supplement figure (Fig. S1). We have also revised the main text to clarify the usage of F-net and the advantage of its depths.

Line 253-314:” *For offshore earthquakes, source depths from the Full Range Seismograph Network of Japan (F-net) catalogue²⁷ which is based on waveform modeling are of better quality, and the focal mechanism information in the catalogue is useful for selecting interplate events (see Supplementary Figure 1 for the consistency of F-net depth and S-P time from S-net). We have selected 287 interplate earthquakes with $M_w \geq 3.5$ based on their focal mechanisms while taking into account their depth information in the catalogue of the F-net²⁷ (see Methods).*”

Comment 6:

3. While it is clear that the splitting time is relatively small near the stagnant wedge corner, it does not always reflect the strength of anisotropy. Another likely scenario is that the symmetry axis is sub-vertical, given the thin wedge corner, it can result in apparent isotropy. As serpentinised mantle wedge is often invoked to interpret a hydrate wedge corner and decoupling, it is probably relevant to discuss the observations with respect to the anisotropy symmetry of serpentine.

Reply: Thank you for this important comment. We have added discussion on the effect of symmetry axis (Lines 501-514). For the forearc area of the Japan trench, large-scale serpentinization of the mantle wedge is not observed (e.g., Tsuji et al., GRL, 2008), with the reason being that slab dehydration does not peak until a much greater depth (Wada and Wang, 2009; Abers et al., Nature Geosci, 2017). Therefore, we did not discuss the effect of

serpentine.

Lines 501-514: *“Anisotropy with a sub-vertical symmetry axis may also be invoked to explain the small splitting, but the consistency of this pattern over the entire ~700km of the margin including the kink between the Kuril and Japan trenches makes this possibility extremely unlikely.”*

Comment 7:

minor comments:

1. line 54-60, the authors illustrate several source areas that may contribute to splitting observations in the fore-arc area. However, the mechanism producing seismic anisotropy discussed in each area is quite limited. Several alternative mechanisms that may account for the observations deserve to be noted. A broader discussion here will also help the authors to examine their result in the context of alternative scenarios. On the other hand, it is not clear if the new observations, while very valuable, can necessarily distinguish alternative mechanisms for a given area. To clarify, the authors identified likely area resulting in observed splitting, but they did not attempt to use their measurement to distinguish alternative mechanism in a given area.

Reply: This comment arose because of our failure to explain the objective of the paper more clearly, the same reason as for Reviewer #1's comment 2. See reply to Reviewer #1's comment above. We have revised the paper to clarify that this study is focused only on the anisotropy of the mantle wedge. For the mantle wedge, the most relevant mechanism for the anisotropy is slab driven flow and consequently preferred mineral orientation (including the B-type olivine).

Comment 8:

2. line 54-60, and line 64-71. These two sections may deserve some rearrangement. Given the local splitting measurement from interpolate and upper plate events, it seems that sub-slab and slab anisotropy do not come into play in this work since they are not sampled. It is unclear the writing is consistent with the focus of this work. Similarly, discussions in line 189-191 seem not relevant.

Reply: Thank you for this suggestion. We have removed this paragraph (see reply to Reviewer #1's comment 9 above). We also changed the title and abstract to clarify the scope of this study. Please also refer to the reply to Comment 2 of reviewer #1.

Comment 9:

3. White dot shown in Fig. 2b can be a bit misleading (e.g., null measurement with zero splitting time). Since there is no upper plate earthquake close to the trench, it is probably better to simply display inter-plate earthquake measurement (no white bar, or white dot).

Reply:

We have removed the white dots.

Comment 10:

4. It will be useful to display the distribution of incident angle at several representative stations.

Reply: We have added Supplementary Figure 2 showing the incident angle for 16 stations. We have also added the following text.

Lines 348-420 “*The splitting parameters at each station generally do not exhibit a dependence on incident angle and back-azimuth (Supplementary Figure 2, Method).*”

Comment 11:

5. In Fig. 3b, fast splitting directions around ~110-170 km from the trench vary more significantly than those in other fore-arc area, but the error bar does not seem to reflect this?

Reply: We used circular standard deviation for the error bars that utilize the direction and length of the summed unit vectors (Mardia and Jupp, 2009). Please note there are 180 degree ambiguity in the direction and -130 degree (smallest value of the vertical axis) and +50 degree (largest value of the vertical axis) means the same value.

Reply to the reviewer 3

Thank you for the review. Below I show your comments in orange and our replies in black. The line number is based on the annotated pdf file.

Comment 1:

The manuscript “Subduction dynamics from forearc anisotropy mapped with S-net seafloor seismometers“ presents evidence for widespread trench parallel fast axes in the crustal forearc of the Japan subduction zone from the systematic analysis of a globally relevant and novel dataset. The authors also make a strong case for limited anisotropy in the stagnant mantle wedge. I believe that this paper will find widespread interest in seismotectonics, geodynamics and geologic communities and I support its publication with minimal revision. This work will also go a long way to constraining geodynamic models of Japanese crustal deformation -- necessary to understanding the seismic hazard consequences of the 2011 great Tohoku earthquake.

The paper is well-written. The language is clear and succinct. My minor grammatical suggestions are tracked in the manuscript. The figures are high quality and easy to understand.

This work is notable and to be credited for its simplicity of observations. The basis for the paper is almost a direct observation. Previous criticisms of isolating layer anisotropy are both its path-integrated nature and the regularization necessary to perform anisotropy tomography. The current project overcomes these issues by brute force and abundance of data.

The two key conclusions, and each individually would warrant publication, are 1) persistent trench parallel fast axes in the upper plate and 2) lack of strong anisotropy in the stagnant mantle wedge.

Reply:

Thank you for these comments.

Comment 2:

Of note, there have previously been other consistent observations for strike-parallel fast axes in the forearc crust, but this is by far the most compelling, owing mostly to the data available for analysis. In fact, Fry et al., (2010) published direct evidence of inherited trench parallel fast directions for European subduction (doi:10.1016/j.epsl.2010.06.008).

Reply:

Thank you for letting us know the paper. We have added the paper as an example of trench parallel crustal anisotropy (reference #13).

Comment 3:

Please see appended additional information for a comparison of those results.

When ascribing the source of crustal anisotropy, please describe “structural fabrics”. Are these embriate structural domains? Pervasive and oriented patterns of crustal deformation? More and more evidence is suggesting that fluids escaping from the slab may travel in trench parallel deformation bands. Perhaps this is a direction for further enquiry with the S-Net data.

Reply:

Structural fabrics refers to the predominant orientation of active faults shown in Fig. 2b. These faults may act as fluid paths and hence further enhance anisotropy. We have revised the sentence in lines 646-647 as follows.

Lines 646-647: *“Fig. 2b shows that the fast axes determined in this work are consistent with the prevalent trench parallel strike direction of active crustal faults³⁵.”*

Comment 4:

Have the authors thought about the competing trench sub-perpendicular fast axes in domains F1 and F3? I would like to see at least mention of these trends.

Reply:

Yes. There are several stations with trench sub-perpendicular fast axes. We think some of the fast axes that are sub-perpendicular to the trench can be explained by regional inhomogeneity in structure and stress. We have mentioned these trends as follows.

Lines 648-650: *Exceptions near the boundary of regions F1 and F2 are consistent with NNW-SSE trending trust faults due to the arc-arc collision in this corner by the sliver motion of the Kuril forearc⁴¹.*

Comment 5

On page 4, “. The remarkable non-uniqueness of the interpretations seen in the literature is to a large part due to limitations in the spatial coverage and reliability of the observations.”,

A bigger part of the problem is the limited vertical resolution (not spatial coverage) of splitting techniques – observations can be viewed as a path-integrated average.:

Reply:

Thank you for pointing out this important point. According to your comments we changed the sentence as follows.

Lines 87-90: *The remarkably poor knowledge of mantle wedge anisotropy is to a large part due to limitations in the spatial coverage of the observations and the vertical resolution of the anisotropy analysis.*

Comment 6

In all cases, I would like to see “fast direction” replaced with “fast axis”, as reciprocity means anisotropy is not a vector. The use of "direction" in the literature is widespread and the authors can continue to do so, but my suggestion is that "axis" is more appropriate.

Reply:

We have replaced “fast direction” with “fast axis” throughout the manuscript.

REVIEWER COMMENTS

Reviewer #1 (Remarks to the Author):

The authors have significantly improved the manuscript and corrected the error in the previous version of Figure 2. I certainly agree with the authors' decision to focus on mantle-wedge dynamics from the very beginning, as also expressed by the new title. The relation to previous findings is much better explained in the current version of the manuscript. In my view, the authors have satisfactorily responded to all concerns raised.

I still have one(minor)comment regarding Fig. 3b: Can the authors explain the large variations in fast axes shown for events at distances between 100 and 200 km distance from the trench? One may expect more scattering for smaller delay times, but between 0 and 100 km distance, the delay times are just as small and the fast axes are more stable.

-even more minor comments

abstract:

34: "dynamics of subduction zone mantle wedge." -> dynamics of the subduction zone mantle wedge.

39: "upper plate crust" -> crust of the upper plate

42 or 43: "features" -> exhibits (?, doubling)

42: "the stagnant part" -> the stagnant, upper part (?)

234: ... "world's" ...

change phrase to: ... the densest systematic mapping of ...forearc to date ...

318: -> catalogue (?)

Reviewer #2 (Remarks to the Author):

I am satisfied with the revision made by the authors with respect to my comments.

I caution on the use of the term "fast axis". One may recognise that splitting measurements refer to observation such as "fast direction", and the term "fast axis" in a way is an interpretation with respect to anisotropy symmetry.

Reviewer #3 (Remarks to the Author):

In my original review, I suggested the paper was a great contribution and suitable for publication with minimal revisions. The authors have addressed all of my concerns sufficiently. Further, the addition of S1 improves the presentation of the results. The rewrite of the introduction also helps to clarify (and properly scale) the papers aims. The revision remains well written and easily understandable.

Reply to the reviewer 1

Thank you for the review. Below I show your comments in orange and our replies in black.

Comment 1:

The authors have significantly improved the manuscript and corrected the error in the previous version of Figure 2. I certainly agree with the authors' decision to focus on mantle-wedge dynamics from the very beginning, as also expressed by the new title. The relation to previous findings is much better explained in the current version of the manuscript. In my view, the authors have satisfactorily responded to all concerns raised.

Reply:

Thank you for the comments.

Comment 2:

I still have one(minor)comment regarding Fig. 3b: Can the authors explain the large variations in fast axes shown for events at distances between 100 and 200 km distance from the trench? One may expect more scattering for smaller delay times, but between 0 and 100 km distance, the delay times are just as small and the fast axes are more stable.

Reply:

We think the variation at distances between 100 and 200 km and 0 and 100 km are similar according to the similar length of blue error bar shown for each distance ranges. Given the much larger number of data (stations) in 100 – 200 km, the outliers are still a minority. The few solid symbols around -120° ($= 60^\circ$) stand out because the vertical range is folded. To clarify the similar standard deviations in 0 -100 and 100-200 km, we revised Figure 3b by removing the green error bar for shallow events which was not discussed and made the blue error bars difficult to read.

Comment 3:

-even more minor comments

abstract:

34: "dynamics of subduction zone mantle wedge." -> dynamics of the subduction zone mantle wedge.

39: "upper plate crust" -> crust of the upper plate

We have change it to "upper-plate crust". [We need to stay within the abstract's word limit]

42 or 43: "features" -> exhibits (?, doubling)

42: "the stagnant part" -> the stagnant, upper part (?)

We have changed it to "the stagnant, most seaward part"

234: ... "world's" ...

change phrase to: ... the densest systematic mapping of ...forearc to date ...

318: catalog -> catalogue (?)

Reply:

We have revised all of these points as suggested or as explained above.

Reply to the reviewer 2

Thank you for the review. Below I show your comments in orange and our replies in black.

Comment 1:

I am satisfied with the revision made by the authors with respect to my comments.

I caution on the use of the term "fast axis". One may recognise that splitting measurements refer to observation such as "fast direction", and the term "fast axis" in a way is an interpretation with respect to anisotropy symmetry.

Reply:

Thank you for your comments. We have replaced "fast axis" with "fast direction". Note that Reviewer #3 prefers "fast axis". We think either direction or axis is fine, although direction sounds slightly better.

Reply to the reviewer 3

Thank you for the review. Below I show your comments in orange and our replies in black.

Comment 1:

In my original review, I suggested the paper was a great contribution and suitable for publication with minimal revisions. The authors have addressed all of my concerns sufficiently. Further, the addition of S1 improves the presentation of the results. The rewrite of the introduction also helps to clarify (and properly scale) the papers aims. The revision remains well written and easily understandable.

Reply:

Thank you for the continuing encouragement.

Reply to the reviewer 1

Thank you for the review. Below I show your comments in orange and our replies in black.

Comment 1:

I am satisfied with the explanation provided by the authors regarding my question about Fig.3. All other points raised in my review have also been addressed.

However, I certainly would not have recommended to change all occurrences of "fast axis" to "fast direction" and I am not sure that this was really the intension of reviewer_2 either.

We all know that it not possible to assign a clear "direction" to the polarization axis of the fast shear wave, as it is generally ambiguous by 180°. I do agree with reviewer_2 that "fast axis" generally (or should) refer(s) to the properties of the anisotropic medium and not the measurement of polarization.

Personally, I would have preferred the term "fast polarization". To avoid an endless discussion on this topic, the authors may state at one point that their use of the term "direction" does not imply that it can be uniquely resolved.

Reply:

We appreciate the fact that each researcher (and reviewer) has a different preference for what words to use to describe the orientation of polarization. "Fast axis" indeed sounds like talking about a mineral, and "fast polarization" sounds like polarization that occurs quickly. We do not think "fast direction" will cause any scientific confusion. It is like when we say the direction of maximum stress σ_1 is 30 degrees from north, no one will be confused because we do not specify it is also 210 degrees. For physical quantities that are bi-directional by default, it is adequate and customary to specify one of the two opposite directions. We worry that further discussion would unnecessarily complicate the simple issue.